# Tetrasubstituted Pyrrole Derivative Mimetics of Protein–Protein Interaction Hot-Spot Residues: A Promising Class of Anticancer Agents Targeting Melanoma Cells

**DOI:** 10.3390/molecules28104161

**Published:** 2023-05-18

**Authors:** Marco Persico, Paola Galatello, Maria Grazia Ferraro, Carlo Irace, Marialuisa Piccolo, Avazbek Abduvakhidov, Oleh Tkachuk, Maria Luisa d’Aulisio Garigliota, Pietro Campiglia, Patrizia Iannece, Michela Varra, Anna Ramunno, Caterina Fattorusso

**Affiliations:** 1Department of Pharmacy, University of Naples “Federico II”, Via D. Montesano 49, 80131 Napoli, NA, Italy; m.persico@unina.it (M.P.); mariagrazia.ferraro@unina.it (M.G.F.); carlo.irace@unina.it (C.I.); marialuisa.piccolo@unina.it (M.P.); avazbek.abduvakhidov@unina.it (A.A.); oleh.tkachuk@unina.it (O.T.); 2Department of Pharmacy, University of Salerno, Via G. Paolo II 132, 84100 Fisciano, SA, Italy; paola.galatello@gmail.com (P.G.); ml.daulisiog@gmail.com (M.L.d.G.); pcampiglia@unisa.it (P.C.); 3Department of Chemistry and Biology, University of Salerno, Via G. Paolo II 132, 84100 Fisciano, SA, Italy; piannece@unisa.it

**Keywords:** protein–protein interactions, peptidomimetics, melanoma, pyrroles, apoptosis

## Abstract

A new series of tetrasubstituted pyrrole derivatives (TSPs) was synthesized based on a previously developed hypothesis on their ability to mimic hydrophobic protein motifs. The resulting new TSPs were endowed with a significant toxicity against human epithelial melanoma A375 cells, showing IC_50_ values ranging from 10 to 27 μM, consistent with the IC_50_ value of the reference compound nutlin-3a (IC_50_ = 15 μM). In particular, compound **10a** (IC_50_ = 10 μM) resulted as both the most soluble and active among the previous and present TSPs. The biological investigation evidenced that the anticancer activity is related to the activation of apoptotic cell-death pathways, supporting our rational design based on the ability of TSPs to interfere with PPI involved in the cell cycle regulation of cancer cells and, in particular, the p53 pathway. A reinvestigation of the TSP pharmacophore by using DFT calculations showed that the three aromatic substituents on the pyrrole core are able to mimic the hydrophobic side chains of the hot-spot residues of parallel and antiparallel coiled coil structures suggesting a possible molecular mechanism of action. A structure–activity relationship (SAR) analysis which includes solubility studies allows us to rationalize the role of the different substituents on the pyrrole core.

## 1. Introduction

Targeting protein–protein interactions (PPIs) involved in the regulation of the cell cycle represents an established approach in the field of anticancer agents [1]. Among PPIs found in the Protein Data Bank (PDB), 62% have an α-helix at the interface, highlighting the importance of this structural element in protein–protein recognition [2,3]. Accordingly, significant efforts have been allocated to the development of non-peptide α-helix mimetics capable of mimicking the spatial arrangement of key residues in α-helix-mediated PPIs. Although the identification of such ligands using conventional drug-discovery processes has resulted in difficulty, some successful examples have been reported [4,5,6,7,8,9] (Figure 1).

The α-helix-mediated PPIs exhibit considerable diversity; however, hydrophobic residues constitute a majority of hot-spot residues at the interfaces between helices. Accordingly, in all the compounds reported in Figure 1, the common structural feature is represented by the presence of three aliphatic and/or aromatic moieties placed on a constrained chemical skeleton in such a way as to reproduce a specific spatial orientation of three hydrophobic residues on a α-helix. The presence of aromatic moieties is generally associated with an increased affinity for the target sequence, probably due to the establishment of π-π or π-alkyl interactions [10]. In particular, compounds **a** and **b** were reported to mimic the i, i + 3, and i + 4 residues, [4,5] while compounds **c** and **d** were reported as mimicking the i, i + 3, and i + 7 residues of helix-based PPI motifs [6,7]. Finally, nutlin-3a (**e** in Figure 1) [8] and compound **f** [9] were proved to mimic the i, i + 4, and i + 7 residues of the p53 ^19^FxxLWxxL^26^ motif responsible for its interaction with MDM2.

In this scenario, we identified a tetrasubstituted pyrrole (TSP) derivative, namely, 4-benzoyl-*N*-[(3-chlorophenyl)methyl]-5-methyl-1-[(4-methylphenyl)methyl]-1H-pyrrole-2-carboxamide (**1**, Figure 2), as the lead structure able to mimic helix-based hydrophobic protein-recognition motifs [11].

Compound **1** showed selective in vitro anti-cancer activity against M14 and A375 melanoma cell lines and a synergistic effect when used in combination with nutlin-3a indicating that they do not compete for the same binding site but rather hit the same molecular pathway, such as the p53 pathway, reducing p53-MDM2 complex formation [11,12].

On these bases, in the present study, in order to expand the structure–activity relationship (SAR) investigation of this promising class of peptidomimetics, we describe the design, synthesis, in vitro activity, and SAR analysis of a second series of TSP analogues (compounds **9a**–**i**, **10a**,**b**; Figure 2), among which we identified compound **10a** as the most active of the previous and present series of TSPs.

## 2. Results and Discussion

### 2.1. Design and Synthesis of the New TSP Derivatives

The information acquired from our first series of TSPs indicated the three aromatic rings X, Y, and Z as pharmacophore groups able to mimic the side chain of three hydrophobic residues of an α-helix-based PPI motif [11]. The substitution of the Z ring with a methyl group led to complete loss of activity as well as the variation of the length of the methylene linker connecting the amide bond to the Y ring or the substitution of this latter with cyclohexane. In addition, some preliminary requirements for the substitution of the aromatic rings X and Y were identified. Bulk electron-withdrawing groups R_1_ such as NO_2_ and CF_3_ led to inactive compounds, while the *meta*-substitution of the Y ring with halogen led to active compounds. 

Starting from these results, we continued the exploration of the TSP skeleton by varying the structure of the reference compound **1** (Table 1). 

Firstly, we continued the exploration of the *para* position of the X ring by varying the nature (electron-withdrawing/electron-donating) and the size of the R_1_ substituent (i.e., Cl, OCH_3_, F, and NH_2_; **9a**–**c**, and **10a**; Table 1). Then, additional combinations of a *para*-substituted X ring and *meta*-substituted Y ring were obtained by modifying **1** at both R_1_ and R_2_ (i.e., R_1_: *p*-Cl and R_2_: *p*-Cl (**9e**); R_1_: H and R_2_: *m*-CF_3_ (**9f**)). 

Moreover, an attempt to investigate the *para* position of the Y ring was performed by shifting the *meta* chlorine atom of **1** to the *para* position through the synthesis of compound **9d** while the Z phenyl ring of **1** was modified by introducing two electron-donating groups at the *para* position (i.e., CH_3_ and OCH_3_; **9h** and **9i**, respectively). 

Additional structural modifications included the exchange of the Z phenyl ring with the methyl group at position 5 (**9g**) as well as the reduction of the carbonyl group at position 4 of the pyrrole ring (W) to a hydroxyl group (**(±)10b**), thus providing a hydrogen bond interaction with the target and, at same time, modifying the geometry of the carbon atom from planar to tetrahedral. 

The synthetic route of the title compounds **9a**–**i** and **10a,b** is outlined in Figure 1. In detail, reaction of (±)-serine methyl ester hydrochloride **2** with *p*-toluenesulfonyl chloride afforded the derivative **3**, which in turn was reacted with di-*tert*-butyl dicarbonate, (Boc)_2_O, and a catalytic amount of 4-dimethylaminopyridine (DMAP) to give the dehydroalanine derivative **4**. Then, the intermediate **4** was reacted with the appropriate β-diketone **5a**–**c**, using Cs_2_CO_3_ as a base, and the crude mixture was treated with 10% TFA in dichloromethane at room temperature to yield the pyrrole derivatives **6a**–**d** (Figure 1). Both **6a** and its regioisomer **6b** were obtained from **5a** [11].

*N*-benzylation of **6a**–**d** with commercially available benzyl bromides afforded the esters **7a**–**i**, which in turn were hydrolyzed with 2M NaOH at reflux. The corresponding acids **8a**–**i** were precipitated by adding 2M HCl and subjected to a coupling reaction with an appropriate amine using 1-hydroxybenzotriazole (HOBT) and O-benzotriazol-1-yl-*N*,*N*,*N*′,*N*′-tetramethyluronium-hexafluorophosphate (HBTU) as coupling reagents in the presence of *N*-methylmorpholine (NMM) in DMF to provide the final compounds **1** [11] and **9a**–**i** (Table 1) in good yields. Compounds **10a** and **(±)10b** were obtained by reduction of **9j** and **1**, respectively, using Fe/NH_4_Cl or NaBH_4_.

Compounds 1-(4-methylphenyl)butane-1,3-dione **5b** and 1-(4-methoxyphenyl)butane-1,3-dione **5c** were synthesized (Scheme S1) by Claisen condensation of 4-methylacetophenone **11a** or 4-methoxyacetophenone **11b** with ethyl acetate in the presence of NaH (60% *w*/*w*, dispersion in mineral oil).

### 2.2. Solubility Assay

The solubility of all compounds was evaluated in a PBS solution (phosphate-buffered saline; Sigma-Aldrich, Merck KGaA, Darmstadt, Germany, pH = 7.4) by means of UV spectroscopy [13].

For each compound, the precipitation point was quantified measuring the increasing of the UV absorption of PBS in the range of 600–800 nm after successive additions of aliquots of a DMSO solution at specific concentration. Because of the absence of bands in the selected UV region, the increase in absorption is due to light scattering caused by the formation of particulate in the PBS. UV absorption data were graphically reported vs. μL of the added DMSO solution (x-axis), and the precipitation point was identified as the intersect point of the bilinear fitted curve. The x coordinate of the intersect point corresponded to the microlitres of DMSO added to PBS (1.4 mL) when the precipitation occurred, from which the concentration of the molecule in PBS at the precipitation point is readily calculated. 

The graphical representation of the obtained data is reported in Figure 3 (**1**, **10a**, and nutlin-3a) and Appendix A (**9a**–**i** and **±10b**), while the calculated solubility values of all the analyzed compounds are reported in Table 2.

Noteworthy is that the present data revealed that the TSP solubility is sensitive to the type and/or concentration of the ions dissolved in the buffered solution. Indeed, the previously calculated solubility of **1** in 20 mM of PBS (pH = 7.4) containing 150 mM of KCl [11] was higher (≤6.8 µM) than that reported herein. This phenomenon could be related, at least in part, to the different ionic charge densities in the two buffered solutions that can affect the hydrophobic interactions between nonpolar solutes [14,15].

For compounds dissolved in DMSO at the same final concentration, the ratio between the calculated volumes of the DMSO solutions at the precipitation points can be directly related to the ratio of the compound solubilities. The newly synthesized compounds **9a**-**i** showed values very close to that found for the reference compounds **1** (3.5 µM) and nutlin-3a (2.4 µM; Table 2). On the contrary, the solubility of compound **10a** resulted in values 5.2- and 2.8-fold higher than that of nutlin-3a and **1**, respectively. Compound **10a** resulted as, indeed, the most soluble among the tested compounds, followed by the racemic mixture of **(±)10b**. The rest of the compounds, including nutlin-3a, were significantly less soluble in the same conditions, with values of solubility ranging from 1.5 to 3.9 µM. 

### 2.3. Biological Evaluation

To evaluate the biological effects of compounds **1**, **9a**–**i**, and **10a**,**b**, we have used specific human tumor models in the context of preclinical investigations. In particular, starting from the results of our previous pharmacological studies [11,12], human malignant cells of different histological origins and endowed with high replicative potential in vitro, such as human melanoma p53wt A375 cells and colorectal carcinoma HCT-116 cells, were used. Moreover, selective toxicity against the healthy cell line L6 was also investigated.

Cell growth inhibition activity of compounds **1**, **9a**–**i**, **10a**,**b**, and the positive control nutlin-3a was evaluated following 48 h of incubation with increasing concentrations of the compounds (from 1.5 to 25 µM) in human cancer (A375 and HCT-116) and healthy (L6) cell lines. Following treatments in vitro for 48 h at the indicated concentrations, IC_50_ values were calculated by concentration/effect curves based on the estimation of a “cell survival index”, a parameter derived concurrently from the evaluation of the cellular metabolic activity and the live/dead cell ratio, as reported in Section 3 (Table 3). The evaluation of the “cell survival index” show a typical concentration-dependent sigmoid trend, achieving IC_50_ values in the low micromolar range (Appendix A).

Overall, as reported in Table 3, all tested compounds resulted as more active against the melanoma A375 cell line with respect to the colon cancer HCT-116 line. Indeed, all compounds showed IC_50_ values in the melanoma A375 cell line ranging from 10 to 27 μM, consistent with the IC_50_ value of nutlin-3a (15 ± 3 µM; Table 3). On the contrary, when the new TSP derivatives were tested against HCT-116 cells, **9i** resulted as inactive (IC_50_ > 100 µM) and the rest of the compounds showed IC_50_ values ranging from 25 to 56 μM, higher than that of nutlin-3a (IC_50_ = 12.5 ± 5 µM; Table 3).

Importantly, concerning the cytotoxicity against the healthy L6 cells, all compounds showed IC_50_ values ranging from 35 to 72 µM with the exception of **9d** (IC_50_ = 25 ± 6 µM). Accordingly, the selectivity index with respect to A375 was between 1.3 and 5.6, thus, it resulted higher than that showed by nutlin-3a (i.e., 1.2; Table 3). 

Going into details, no significant difference in IC_50_ values of the A375 cell line were observed among the newly synthesized pyrrole derivatives compared to lead **1**, with the exception of (i) **9c** (IC_50_ = 13 ± 4 µM), bearing as R_1_ a strong electron-withdrawing and a potential hydrogen bond acceptor substituent such as the fluorine atom, and (ii) **10a** (IC_50_ = 10 ± 4 µM), bearing an electron-donating and hydrogen bond donor/acceptor substituent (i.e., the amino group) at the same position (Table 1 and Table 3). In particular, compound **10a** was the most active pyrrole derivative among the tested compounds, resulting as slightly more active than nutlin-3a (Table 3). 

Taken together, these results suggest a mechanism of action for the new TSPs similar to that of the reference compound **1** based on the interference with a specific molecular pathway, leading to apoptosis, and not dependent on an unspecific cytotoxicity.

### 2.4. Apoptosis Studies

To support our hypothesis that the mechanism of action of these TSP derivatives is related to their ability to interfere with PPIs involved in the cell cycle regulation of cancer cells, the most active compounds against both the melanoma A375 cell line (i.e., **10a** and **9c**) and colon cancer HCT-116 line (i.e., **9f** and **(±)10b**) were selected to further investigate their mechanism of action. Since cell death typically occurs by apoptosis, necrosis, or autophagy, in the first instance we evaluated their ability to induce cellular morphological changes in A375 cell lines after exposure for 48 h at IC_50_ micromolar concentrations of selected compounds (for **9c** and **10a**, IC_50_ 13 ± 4 and 10 ± 4 µM, respectively), and the obtained results were compared with those of nutlin-3a, used as a positive control (Figure 4).

Cytomorphological analyses by phase-contrast light microscopy for the dynamic cell population monitoring highlighted important modifications, suggestive of the possible activation of programmed cell-death pathways (Figure 4). In particular, microscopy analyses provided evidence that reduction in cell survival index is associated with well-detectable cytotoxic effects and distinctive morphological hallmarks of apoptosis. Indeed, apoptosis onset is characterized by membrane blebs and cell shrinkage, and culminates in the formation of balloon-like structures indicating the loss of plasma membrane integrity. Besides losing their normal morphological features, the rounding up of the cells visibly increased after 48 h of incubation, with an enhancement of the surface blebbing and cell shrinkage.

In order to confirm the possible activation of apoptosis, we performed fluorescence experiments to simultaneously monitor apoptotic and/or necrotic cells as well as deepen the biological effects on A375 triggered by exposure to the most bioactive derivatives under investigation (Figure 5).

As clearly emerges from fluorescence micrographs by using the phosphatidylserine (PS) sensor (green fluorescence, FITC filter), a consistent activation of apoptosis was observed after incubation with IC_50_ concentrations of **9c** and **10a**. Indeed, in apoptosis, PS is transferred to the outer leaflet of the plasma membrane. As a general indicator of the initial/intermediate stages of programmed cell death, the PS appearance on the cell surface can be detected as an apoptotic hallmark. Even in this circumstance, the results are very similar to those obtained with nutlin-3a. 

Finally, concurrent necrotic phenomena in the same experimental conditions were excluded (red fluorescence, TRITC filter; Figure 5).

Taken together, these results indicate that the anticancer activity of the new TSP derivatives, as in the reference compound **1** [12], is related to their ability to activate apoptotic cell-death pathways.

### 2.5. Computational Studies 

#### 2.5.1. Conformational Analysis

The conformational properties of **9a**–**i**, **10a**, and **(±)-10b** were calculated through an in-depth conformational analysis integrating force-field-based and quantum mechanical computational methods. In order to estimate the effect of the introduced structural modifications on the conformational properties of the new TSPs, the former lead **1** was included in the analysis. Indeed, the conformational space of the first series of TSPs was explored by applying in vacuum semi-empirical PM6 [16] calculations [11] while, in the present study, we performed density functional theory (DFT) [17] calculations and used the conductor-like polarizable continuum model (C-PCM) [18] to mimic an aqueous environment.

Results showed two parallel sets of specular conformers for each compound, which are characterized by the same conformational energy and the same absolute value of the torsion angles but with the opposite sign (i.e., conformational enantiomers, named E1 and E2; Table 4 and Appendix A; Appendix A). In the case of the chiral compound **10b**, tested as a racemic mixture, the specular conformers corresponded to those of the enantiomer with the opposite configuration. 

The low-energy (ΔE_GM_ < 2 kcal/mol) conformers of **1**, **9a**–**i**, and **10a** can be clustered into two conformational families named TCC and TCT according to the antiperiplanar (T) or synperiplanar (C) conformation of the torsion angles τ1, τ2, and τ3 (Appendix A; Table 4). The TCC family is favored over the TCT family in terms of both conformer population (~28% vs. ~25%) and conformational energy.

In compound **(±)-10b**, the carbonyl group at C4 is reduced to a hydroxyl function, thus changing the hybridization of the τ3 torsion angle from sp^2^ to sp^3^. Compound (±)-**10b** reproduced the same conformational preference as was observed for **1**, **9a**–**i**, and **10a** concerning the torsional angles τ1 and τ2 (family TC) while presenting three different orientations of the Z ring (i.e., three values of τ3 corresponding to the G^−^, A^−^, and G^+^; Appendix A; Figure 6D and Appendix A).

Finally, the low energy conformers of **9g** showed values of τ1 and τ2 corresponding to the conformational families TC (most favored also in this case) and TT. In compound **9g**, the Z ring is moved from the carbonyl group at C4 (whose orientation is defined by τ3) to C5. In this case, the τ3 torsion angle always presents a synperiplanar (C) conformation with respect to C5 (Appendix A; Figure 6C and Appendix A).

Noteworthy is that all compounds presented not only the same conformational preference concerning τ1 and τ2 torsion angles (TC family) but also similar values of τ4 (~90°). This is likely due to the establishment of an intra-molecular hydrogen bond between the carbonyl group of the amide function and the X ring (Figure 6 and Appendix A). 

On the other hand, four possible conformations of τ5 were found: (i) A^−^ (−90° to −150°; subfamily I), (ii) G^−^ (30° to 90°; subfamily II), (iii) A^+^ (90° to 150°; subfamily III), and T (±150° to 180°; subfamily IV) (Appendix A; Table 4 and Appendix A).

In summary, the low-energy conformers of the new TSPs as well as those of the former lead **1** can be classified in two conformational families, which are characterized by similar values of τ1, τ2, and τ4 while differing in the values of τ3 and τ5.

The comparison of the present with the previous [11] results evidenced some important differences. Indeed, (i) the previous PM6 low-energy conformers included the *cis* conformation of the amide bond (τ1) and (ii) those belonging to the TCC and TCT families showed a decreased distance between the X and Y rings compared to DFT (Appendix A), likely due to the overestimation of intra-molecular π-π interactions in the absence of any solvent.

#### 2.5.2. Investigation of the Peptidomimetic Ability of the New TSPs

In order to identify the helix-based hydrophobic motif(s) whose consensus structure (i.e., the spatial orientation of the hot-spot residues) can be mimicked by TSPs, the peptidomimetic ability of **1**, **9a**–**i**, **10a**, and **(±)-10b** was investigated by performing a comparison of their low-energy DFT conformers with the experimentally determined structures of hydrophobic PPI motifs (coiled coil and LxxLL). 

In particular, the TSP conformers were grouped according to their intra-molecular distances (Appendix A), the PPI hot-spot residues were identified, and the distances between the centroids of the interacting side chains (Appendix A) were compared with the distances between the centroids of the X, Y, and Z rings (Table 4 and Appendix A). The TSP conformers matching specified distance similarity criteria with the PPI hot-spot residues (described in Section 3) were superimposed on the PPI motifs by fitting the centroids of the aromatic rings X, Y, and Z on the centroids of the side chains of the hot-spot residues (Appendix A). The conformers with a calculated root-mean-square distance (RMSD) value < 2.5 Å were selected as mimetic of the PPI motif. 

Four different pharmacophores were identified (corresponding to the conformers reported in Figure 6) able to reproduce the intra-molecular distances between the hot-spot residues of hydrophobic PPI motif involved in the formation of parallel and antiparallel coiled coil structures (Table 5). 

Interestingly, one result is that the orientation of the TSP pharmacophore moieties (i.e., X, Y, and Z rings) and the presence of conformational enantiomers reflect some structural properties of PPI motifs involved in the formation of parallel and antiparallel coiled coil structures.

The i *+* 3 and i + 4 hot-spot residues of the parallel and antiparallel coiled coil structures are symmetrically positioned with respect to the helix axis and to the other hot-spot residues i and i + 7, thus allowing the recognition of two different faces of the α-helix in the two types of PPI (parallel and antiparallel coiled coil structures; Figure 7).

A similar structural relationship is present in the conformational/configurational enantiomers of the TSP derivatives (Figure 8), and this peculiarity may contribute to their peptidomimetic ability. The distance between the Y and Z rings represents the longest intra-molecular distance (~10 Å; Figure 6A–C and Appendix A) and the X ring mimics the “central” hot-spot residue of the motif (i.e., i + 3 or the i + 4 in parallel or antiparallel coiled coil structure, respectively). The Y and Z rings can alternately mimic the hot-spot residue i or i + 7 depending (a) on the enantiomer when comparing the two enantiomers to the same PPI motif (Figure 8A vs. Figure 8B and Figure 8C vs. Figure 8D) or (b) on the PPI motif when comparing the same enantiomer to the two motifs (Figure 8A vs. Figure 8C and Figure 8B vs. Figure 8D). 

Only in the case of one pharmacophore of **(±)-10b** (n.4 in Table 5), the distance between the X and Y ring is the longest (~10 Å; Figure 6D). By consequence, the Z ring mimics the “central” hot-spot residue of the motif and the X and Y can alternately mimic the hot-spot residue i or i *+* 7. Pharmacological studies evidencing that the anticancer activity of TSPs is related to the activation of apoptotic cell-death pathways, confirming the mechanism of action of compound **1** [12] and in line with computational results, suggested the same mechanism of actions for the new TSPs. The pharmacological investigation previously performed on the effect of **1** on melanoma cells (M14 [11] and A375 [12]) evidenced its ability to impair melanoma viability reducing p53-MDM2 interaction and restoring p53 functions and transcriptional activities. Melanoma is one of the few types of cancer in which p53 is not frequently mutated; however, p53 inactivation can be indirectly achieved by a stable activation of MDM2 induced by a deletion in CDKN2A (Cyclin-Dependent Kinase Inhibitor 2A) locus encoding for p14^ARF^ [19].

p14^ARF^ is a tumor suppressor that regulates p53 function through the inhibition of the formation of the p53-MDM2 complex (that leads to p53 degradation). In particular, its 37 N-terminal amino acids bind (to a different region from that necessary for the binding of p53) and sequester MDM2 within nucleoli, causing cell cycle arrest [20]. 

Based on the hypothesis that the TSP derivatives could mimic p14^ARF^ in the interaction with MDM2, we performed a bioinformatics and structural analysis on the first two N-terminal helices of p14^ARF^ (αH1 (4–14) and αH2 (20–29)) by using the x-ray structure of the mouse p19^ARF^ tumor suppressor protein (PDB ID: 1HN3 (Appendix A). 

Interestingly, we found on the αH2 helix of p14^ARF^ a coiled-coil-like motif characterized by the presence of hydrophobic residues at position i, i + 3, i + 4, and i + 7 (i.e., ^220^VRVFVVHI^27^). Noteworthy is that the coiled-coil-like motif predicted on the αH2 helix of p14^ARF^ is part of a region involved in the interaction with MDM2, which acts as a negative allosteric modulator for the binding of p53 to MDM2 [20]. Thus, it can be hypothesized that the TSP derivatives reproducing some hot-spot residues of the interactions of p14^ARF^ with MDM2 could bind to the latter, leading to a decrease in the formation of the p53-MDM2 complex (Figure 9; Appendix A).

However, no structural data are available on the p14^ARF^/MDM2 complex; therefore, the modeling of the interaction of TSPs with MDM2 is not trivial and deserves future dedicated studies.

### 2.6. SAR Analysis

Once it was given that all the compounds share the ability to reproduce (with the X, Y, and Z ring) the spatial orientation of the side chain of the i, i + 3/4, and i + 7 residues of a coiled coil PPI motif, we tried to rationalize the observed SARs by separating the effect of the pharmacokinetic and pharmacodynamic properties of the compounds on the biological activity.

The cLogD of nutlin-3a, **1**, and the new TSP derivatives **9a**–**i** and **10a,b** was calculated and resulting values were about 5 or higher (Appendix A), indicating a high lipophilicity of the compounds as evidenced by the results of solubility assays.

In light of the obtained results, some considerations can be made: (i) a large number of compounds show the same solubility but different activity, and (ii) although no relationships were found between compound activity and solubility (R^2^ = 0.2963), the two compounds nevertheless show significantly higher solubility; **(±)-10b** and **10a** are the most active ones. Going into detail, the effect of the introduction of the NH_2_ group as R_1_ (**10a**) as well as the reduction of the carbonyl group (**(±)-10b**) increased both solubility and activity. Accordingly, it cannot be excluded that, in these cases, pharmacokinetic effects played a role in the observed increase in activity. On the contrary, the substitution of the CH_3_ group with the fluorine atom in the *para* position of the X ring (**9c**), the shift of chlorine atom from the *meta* to the *para* position on the Y ring (**9d**), and the introduction of a methoxy group in the *para* position of the Z ring (**9i**) led to an increase in the activity not related to an increase in solubility. The same is valid when the Z ring is exchanged with the methyl group at C5 (**9g**) and the activity is retained despite a decreased solubility (Figure 10). 

## 3. Materials and Methods

### 3.1. Chemistry

Melting points were measured on a Gallenkamp melting point apparatus. NMR spectra were acquired on a Bruker Avance 300 MHz or a Bruker Ascend 400 MHz spectrometer. Splitting patterns are described as singlet (s), doublet (d), triplet (t), quartet (q), quintuplet (qt), and broad (br). Mass spectra were obtained by electrospray ionization (ESI-HRMS, positive mode) using an LTQ Orbitrap XL mass spectrometer (Thermo Fisher Scientific, San Jose, CA, USA) equipped with Xcalibur software. The Orbitrap mass analyzer was calibrated according to the manufacturer’s directions using a mixture of caffeine, methionine, arginine, phenylalanine, and alanine acetate (MRFA) and Ultramark 1621 in a solution of acetonitrile, methanol, and acetic acid. Chromatographic separations were performed on silica gel (Kieselgel 40, 0.040–0.063 mm, Merck KGaA, Darmstadt, Germany). Reactions and product mixtures were routinely monitored by thin-layer chromatography (TLC) on Merck 0.2 mm precoated silica (60 F254) aluminum sheets with visualization by irradiation with a UV lamp. All starting materials, reagents, and solvents (reagent-grade) were purchased from Sigma-Aldrich and used without further purification. Compounds **1**, **3**, **4**, **6a,b**, **7a**, **7e**, **7f**, **8a**, **8e**, **8f** and **9j** were obtained as previously reported [11].

*Methyl 5-methyl-4-(4-methylbenzoyl)-1H-pyrrole-2-carboxylate* (**6c**) 

To a suspension of methyl 2-(*N*-(*tert*-butoxycarbonyl)-4-methylphenyl-sulfonamido)acrylate **4** (0.925 g, 2.60 mmol) in dry CH_3_CN (20 mL), 1-(*p*-tolyl)butane-1,3-dione **5b** (0.504 g, 2.86 mmol) and Cs_2_CO_3_ (1.27 g, 3.90 mmol) were added, and the resulting mixture was stirred at room temperature for 12 h. Then, the solvent was removed under reduced pressure, and the resulting residue was dissolved in EtOAc and washed with H_2_O. The organic phase was dried (Na_2_SO_4_) and filtered, and the solvent was removed under vacuum. The resulting residue was dissolved in DCM (18 mL), and trifluoroacetic acid (TFA) (2.0 mL) was added. After 12 h, the organic phase was washed with brine (2 × 25 mL) and 1 M NaOH (2 × 20 mL), dried (Na_2_SO_4_), and filtered, and the solvent was removed under vacuum. The resulting residue was purified by flash column chromatography (SiO_2_, EtOAc/petroleum ether, 40–60 °C, 1:1 *v*/*v*) to provide the title compound as a white powder. Yield: 65%. ^1^H NMR (300 MHz, CDCl_3_) δ: 2.45 (s, 3H), 2.65 (s, 3H), 3.87 (s, 3H), 7.11 (s, 1H), 7.28 (d, 2H, *J* = 7.6 Hz), 7.73 (d, 2H, *J* = 7.9 Hz), 9.56 (brs, 1H).

*Methyl 4-(4-methoxybenzoyl)-5-methyl-1H-pyrrole-2-carboxylate* (**6d**) 

Starting from 1-(4-methoxyphenyl)butane-1,3-dione **5c** and following the same procedure described for **6c**, the title compound was obtained as a white powder. Yield: 58%. ^1^H NMR (300 MHz, CDCl_3_) δ: 2.64 (s, 3H), 3.88 (s, 3H), 3.90 (s, 3H), 6.98 (d, 2H, *J* = 8.7 Hz), 7.12 (s, 1H), 7.85 (d, 2H, *J* = 8.7 Hz), 9.52 (brs, 1H).

*General Procedure for the Synthesis of Esters* (**7a**–**i**)

Sodium hydroxide (8 mL, 50% *w*/*v*) and tetra-*n*-butylammonium hydroxide (TBAH) (0.5 mL, 40% *w*/*v*) were added to a stirred mixture of esters **6a**–**d** (1 mmol) in dichloromethane (12 mL) at 0 °C. After 15 min, a solution of the appropriate aryl bromide (1.5 mmol) in dichloromethane (5 mL) was added dropwise. The resulting mixture was allowed to warm to room temperature, stirred in these conditions for 18h, and then washed with H_2_O, 2M HCl, and brine. The organic phase was dried (Na_2_SO_4_) and filtered, and the solvent was removed under vacuum. The resulting residue was purified by flash chromatography (SiO_2_, EtOAc/*n*-hexane, 1:4 *v*/*v*).

*Methyl 4-benzoyl-1-(4-chlorobenzyl)-5-methyl-1H-pyrrole-2-carboxylate* (**7b**) 

White powder. Yield: 67%. ^1^H NMR (300 MHz, CDCl_3_) δ: 2.58 (s, 3H), 3.77 (s, 3H), 5.69 (s, 2H), 6.96 (d, 2H, *J* = 8.2 Hz), 7.26–7.32 (m, 3H), 7.48–7.61 (m, 3H), 7.83 (d, 2H, *J* = 7.1 Hz).

*Methyl 4-benzoyl-1-(4-methoxybenzyl)-5-methyl-1H-pyrrole-2-carboxylate* (**7c**) 

White powder. Yield: 74%. ^1^H NMR (300 MHz, CDCl_3_) δ: 2.62 (s, 3H), 3.80 (s, 3H), 3.82 (s, 3H), 5.68 (s, 2H), 6.88 (d, 2H, *J* = 8.7 Hz), 7.00 (d, 2H, *J* = 8.7 Hz), 7.26 (s, 1H), 7.49–7.61 (m, 3H), 7.84 (d, 2H, *J* = 8.4 Hz).

*Methyl 4-benzoyl-1-(4-fluorobenzyl)-5-methyl-1H-pyrrole-2-carboxylate* (**7d**) 

White solid. Yield: 66%, mp: 87–89 °C. ^1^H NMR (300 MHz, CDCl_3_) δ: 2.58 (s, 3H), 3.77 (s, 3H), 5.68 (s, 2H), 7.01 (m, 4H), 7.26 (s, 1H), 7.46–7.57 (m, 3H), 7.82 (d, 2H, *J* = 6.8 Hz).

*Methyl-4-acetyl-1-(4-methylbenzyl)-5-phenyl-1H-pyrrole-2-carboxylate* (**7g**) 

White solid. Yield: 77%, mp: 116–118 °C. ^1^H NMR (300 MHz, CDCl_3_) δ: 2.13 (s, 3H), 2.30 (s, 3H), 3.81 (s, 3H), 5.39 (s, 2H), 6.68 (d, 2H, *J* = 8.0 Hz), 7.03 (d, 2H, *J* = 8.0 Hz), 7.23 (d, 2H, *J* = 8.0 Hz), 7.37–7.47 (m, 3H), 7.58 (s, 1H).

*Methyl 5-methyl-4-(4-methylbenzoyl)-1-(4-methylbenzyl)-1H-pyrrole-2-carboxylate* (**7h**) 

White solid. Yield: 73%, mp: 110–112 °C. ^1^H NMR (300 MHz, CDCl_3_) δ: 2.33 (s, 3H), 2.45 (s, 3H), 2.57 (s, 3H), 3.77 (s, 3H), 5.68 (s, 2H), 6.91 (d, 2H, *J* = 7.5 Hz), 7.13 (d, 2H, *J* = 7.5 Hz), 7.26–7.31 (m, 3H), 7.75 (d, 2H, *J* = 7.8 Hz).

*Methyl 4-(4-methoxybenzoyl)-5-methyl-1-(4-methylbenzyl)-1H-pyrrole-2-carboxylate* (**7i**) 

White solid. Yield: 79%, mp: 129–132 °C. ^1^H NMR (300 MHz, CDCl_3_) δ: 2.33 (s, 3H), 2.54 (s, 3H), 3.77 (s, 3H), 3.90 (s, 3H), 5.67 (s, 2H), 6.91 (d, 2H, *J* = 7.5 Hz), 6.98 (d, 2H, *J* = 7.9 Hz), 7.13 (d, 2H, *J* = 7.8 Hz), 7.27 (s, 1H), 7.86 (d, 2H, *J* = 7.2 Hz).

*General Procedure for the Synthesis of Acids* (**8a**–**i**)

To a solution of esters **7a**–**i** (1 mmol) in methanol (10 mL), 2M NaOH (5 mmol) was added, and the resulting mixture was refluxed until no starting material was detected by TLC (DCM as eluent). The solvent was removed under reduced pressure; then, the residue was acidified with 2M HCl (pH ~ 3) and extracted with EtOAc. The organic phase was dried (Na_2_SO_4_) and filtered, and the solvent was removed under vacuum. The resulting residue was purified by flash chromatography (SiO_2_, EtOAc/EtOH, 9:1 *v*/*v*, as eluent) and recrystallized from ethyl ether/petroleum ether (40–60 °C).

*4-Benzoyl-1-(4-chlorobenzyl)-5-methyl-1H-pyrrole-2-carboxylic acid* (**8b**) 

White solid. Yield: 95%, mp: 163–165 °C. ^1^H NMR (300 MHz, DMSO*d*_6_) δ: 2.45 (s, 3H), 5.73 (s, 2H), 7.02 (s, 1H), 7.05 (s, 2H), 7.40 (d, 2H, *J* = 8.4 Hz), 7.52–7.65 (m, 3H), 7.72 (d, 2H, *J* = 6.8 Hz).

*4-Benzoyl-1-(4-methoxybenzyl)-5-methyl-1H-pyrrole-2-carboxylic acid* (**8c**) 

White solid. Yield: 97%. ^1^H NMR (300 MHz, CDCl_3_) δ: 2.61 (s, 3H), 3.81 (s, 3H), 5.64 (s, 2H), 6.87 (d, 2H, *J* = 8.6 Hz), 6.99 (d, 2H, *J* = 8.5 Hz), 7.40 (s, 1H), 7.48–7.61 (m, 3H), 7.84 (d, 2H, *J* = 7.2 Hz).

*4-Benzoyl-1-(4-fluorobenzyl)-5-methyl-1H-pyrrole-2-carboxylic acid* (**8d**)

White solid. Yield: 96%, mp: 187–188 °C. ^1^H NMR (300 MHz, DMSO*d*_6_) δ: 2.50 (s, 3H), 5.73 (s, 2H), 7.07 (m, 3H), 7.18 (m, 2H), 7.51–7.64 (m, 3H), 7.72 (d, 2H, *J* = 7.2 Hz).

*4-Acetyl-1-(4-methylbenzyl)-5-phenyl-1H-pyrrole-2-carboxylic acid* (**8g**)

White solid. Yield: 78%, mp: 173–175 °C. ^1^H NMR (300 MHz, CDCl_3_) δ: 2.14 (s, 3H), 2.30 (s, 3H), 5.37 (s, 2H), 6.68 (d, 2H, *J* = 7.7 Hz), 7.03 (d, 2H, *J* = 7.7 Hz), 7.23 (d, 2H, *J* = 7.1 Hz), 7.38–7.49 (m, 3H), 7.69 (s, 1H).

*5-Methyl-4-(4-methylbenzoyl)-1-(4-methylbenzyl)-1H-pyrrole-2-carboxylic acid* (**8h**) 

White solid. Yield: 93%, mp: 193–195 °C. ^1^H NMR (300 MHz, CDCl_3_) δ: 2.33 (s, 3H), 2.45 (s, 3H), 2.56 (s, 3H), 5.65 (s, 2H), 6.91 (d, 2H, *J* = 7.2 Hz), 7.13 (d, 2H, *J* = 7.2 Hz), 7.28 (d, 2H, *J* = 7.5 Hz), 7.38 (s, 1H), 7.74 (d, 2H, *J* = 7.8 Hz).

*4-(4-Methoxybenzoyl)-5-methyl-1-(4-methylbenzyl)-1H-pyrrole-2-carboxylic acid* (**8i**)

White solid. Yield: 93%, mp: 202–204 °C. ^1^H NMR (300 MHz, CDCl_3_) δ: 2.33 (s, 3H), 2.54 (s, 3H), 3.90 (s, 3H), 5.65 (s, 2H), 6.91 (d, 2H, *J* = 7.6 Hz), 6.98 (d, 2H, *J* = 8.4 Hz), 7.12 (d, 2H, *J* = 7.7 Hz), 7.39 (s, 1H), 7.85 (d, 2H, *J* = 8.4 Hz).

*General Procedure for the Synthesis of Compounds* (**9a**–**i**)

To a solution of the acids **8a**–**i** (1 mmol) in DMF (5 mL), HOBT (2.5 mmol) and HBTU (2.5 mmol) were added, and the resulting mixture was stirred for 20 min at room temperature. Then, *N*-methylmorpholine (NMM) (5 mmol) and the appropriate amine (2.5 mmol) were added, and the resulting mixture was stirred at room temperature for 12 h. The solvent was removed under reduced pressure, and the resulting residue was taken up in EtOAc and washed successively with 2M KHSO_4_, saturated solution of NaHCO_3_, and brine. The organic phase was dried (Na_2_SO_4_), filtered, and concentrated under reduced pressure. The residue was purified by flash chromatography (SiO_2_, EtOAc/petroleum ether (40–60 °C) 1:1 *v*/*v* as eluent) and recrystallized from EtOAc/*n*-hexane to provide the title compounds **9a**–**i** in good yields.

*4-Benzoyl-N-(3-chlorobenzyl)-1-(4-chlorobenzyl)-5-methyl-1H-pyrrole-2-carboxamide* (**9a**) 

White solid. Yield: 89%, mp: 117–119 °C. ^1^H NMR (400 MHz, CDCl_3_) δ: 2.57 (s, 3H), 4.51(d, 2H, *J* = 6.0 Hz), 5.75 (s, 2H), 6.27 (brt, 1H, *J* = 3.4 Hz), 6.88 (s, 1H), 7.02 (d, 2H, *J* = 8.4 Hz), 7.10 (d, 1H, *J* = 6.0 Hz), 7.25–7.34 (m, 5H), 7.49–7.53 (m, 2H), 7.57–7.61 (m, 1H), 7.82 (d, 2H, *J* = 8.4 Hz); ^13^C NMR (101 MHz, CDCl_3_) δ: 12.2, 43.1, 47.9, 115.6, 120.5, 124.8, 126.0, 128.0 (2C), 128.1, 128.6, 129.3, 129.4, 130.3, 132.0, 133.6, 134.9, 136.1, 140.2, 140.6, 142.0, 161.6, 192.3. HRMS (ESI, *m*/*z*) [M + H]^+^ calcd. for [C_27_H_23_Cl_2_N_2_O_2_]^+^ 477.1131; found 477.1126.

*4-Benzoyl-N-(3-chlorobenzyl)-1-(4-methoxybenzyl)-5-methyl-1H-pyrrole-2-carboxamide* (**9b**) 

White solid. Yield: 81%, mp: 137–138 °C. ^1^H NMR (300 MHz, CDCl_3_) δ: 2.56 (s, 3H), 3.79 (s, 3H), 4.48 (d, 2H, *J* = 5.9 Hz), 5.66 (s, 2H), 6.15 (brt, 1H, *J* = 5.4 Hz), 6.80 (s, 1H), 6.84 (d, 2H, *J* = 7.6 Hz), 7.00 (d, 2H, *J* = 8.3 Hz), 7.07 (d, 1H, *J* = 4.8 Hz), 7.19–7.27 (m, 3H), 7.44–7.57 (m, 3H), 7.77 (d, 2H, *J* = 7.9 Hz); ^13^C NMR (75 MHz, CDCl_3_) δ: 12.3, 43.1, 47.9, 55.6, 114.5, 115.6, 120.3, 124.9, 126.1, 128.0, 128.1 (2C), 128.6, 129.4, 129.6, 130.3, 131.9, 134.9, 140.4, 140.7, 142.2, 159.2, 161.8, 192.3. HRMS (ESI, *m*/*z*) [M + H]^+^ calcd. for [C_28_H_26_ClN_2_O_3_]^+^ 473.1626; found 473.1623.

*4-Benzoyl-N-(3-chlorobenzyl)-1-(4-fluorobenzyl)-5-methyl-1H-pyrrole-2-carboxamide* (**9c**) 

White solid. Yield: 69%, mp: 130–132 °C. ^1^H NMR (300 MHz, CDCl_3_) δ: 2.55 (s, 3H), 4.49 (d, 2H, *J* = 6.0 Hz), 5.71 (s, 2H), 6.25 (brt, 1H, *J* = 5.4 Hz), 6.85 (s, 1H), 6.99–7.10 (m, 5H), 7.21–7.29 (m, 3H), 7.45–7.59 (m, 3H), 7.77–7.81 (m, 2H); ^13^C NMR (75 MHz, CDCl_3_) δ: 12.2, 43.0, 47.9, 115.6, 116.0 (d, *J*_CF_ = 21.6 Hz), 120.4, 124.7, 126.0, 128.0, 128.1, 128.3 (d, *J*_CF_ = 8.1 Hz), 128.6, 129.4, 130.3, 132.0, 133.3 (d, *J*_CF_ = 3.2 Hz), 134.9, 140.2, 140.6, 142.1, 161.6, 162.4 (d, *J*_CF_ = 244.5 Hz), 192.3. HRMS (ESI, *m*/*z*)) [M + H]^+^ calcd. for [C_27_H_23_ClFN_2_O_2_]^+^ 461.1427; found 461.1422.

*4-Benzoyl-N-(4-chlorobenzyl)-5-methyl-1-(4-methylbenzyl)-1H-pyrrole-2-carboxamide* (**9d**) 

White solid. Yield: 90%, mp: 154–156 °C. ^1^H NMR (300 MHz, CDCl_3_) δ: 2.37 (s, 3H), 2.58 (s, 3H), 4.49 (d, 2H, *J* = 5.9 Hz), 5.72 (s, 2H), 6.15 (brt, 1H, *J* = 5.6 Hz), 6.82 (s, 1H), 6.95 (d, 2H, *J* = 7.9 Hz), 7.15 (m, 4H), 7.28–7.30 (m, 2H), 7.47–7.60 (m, 3H), 7.81 (d, 2H, *J* = 6.9 Hz); ^13^C NMR (75 MHz, CDCl_3_) δ: 12.2, 21.4, 42.9, 48.2, 115.4, 120.3, 125.0, 126.5, 128.6, 129.1, 129.3, 129.4, 129.8, 131.9, 133.6, 134.6, 137.1, 137.3, 140.4, 142.2, 161.8, 192.4. HRMS (ESI, *m*/*z*) [M + H]^+^ calcd. for [C_28_H_26_ClN_2_O_2_]^+^ 457.1677; found 457.1669.

*4-Benzoyl-N,1-bis(4-chlorobenzyl)-5-methyl-1H-pyrrole-2-carboxamide* (**9e**) 

White solid. Yield: 68%, mp: 147–149 °C. ^1^H NMR (300 MHz, CDCl_3_) δ: 2.52 (s, 3H), 4.44 (d, 2H, *J* = 5.9 Hz), 5.68 (s, 2H), 6.14 (brt, 1H, *J* = 5.4 Hz), 6.80 (s, 1H), 6.96 (d, 2H, *J* = 8.4 Hz), 7.10 (d, 2H, *J* = 8.4 Hz), 7.24–7.29 (m, 4H), 7.43–7.57 (m, 3H), 7.78 (m, 2H); ^13^C NMR (75 MHz, CDCl_3_) δ: 12.1, 42.9, 47.9, 115.5, 120.5, 124.8, 128.0, 128.6, 129.2, 129.3 (2C), 129.4, 132.0, 133.6, 133.7, 136.1, 137.0, 140.2, 142.0, 161.6, 192.3. HRMS (ESI, *m*/*z*) [M + H]^+^ calcd. for [C_27_H_23_Cl_2_N_2_O_2_]^+^ 477.1131; found 477.1122.

*4-Benzoyl-1-benzyl-5-methyl-N-(3-(trifluoromethyl)-benzyl)-1H-pyrrole-2-carboxamide* (**9f**)

White solid. Yield: 87%, mp: 139–141 °C. ^1^H NMR (400 MHz, CDCl_3_) δ: 2.53 (s, 3H), 4.55 (d, 2H, *J* = 6.1 Hz), 5.74 (s, 2H), 6.21 (brt, 1H, *J* = 5.5 Hz), 6.83 (s, 1H), 7.02 (d, 2H, *J* = 6.5 Hz), 7.23–7.57 (m, 10H), 7.77 (m, 2H); ^13^C NMR DEPT-q (101 MHz, CDCl_3_) δ: 12.2, 43.1, 48.5, 115.7, 120.3, 124.3 (q, *J*_CF_ = 273.7 Hz), 124.6 (m, 2C), 124.9, 126.5, 127.7, 128.6, 129.1, 129.4, 129.5, 131.2 (q, *J*_CF_ = 32.3 Hz), 131.9, 137.5, 139.7, 140.3, 142.3, 161.8, 192.3. HRMS (ESI, *m*/*z*) [M + H]^+^ calcd. for [C_28_H_24_F_3_N_2_O_2_]^+^ 477.1784; found 477.1767.

*4-acetyl-N-(3-bromobenzyl)-1-(4-methylbenzyl)-5-phenyl-1H-pyrrole-2-carboxamide* (**9g**) 

White solid. Yield: 74%, mp: 157–159 °C. ^1^H NMR (300 MHz, CDCl_3_) δ: 2.13 (s, 3H), 2.48 (s, 3H), 4.67 (d, 2H, *J* = 5.8 Hz), 5.59 (s, 2H), 6.76 (brt, 1H, *J* = 5.8 Hz), 6.87 (d, 2H, *J* = 7.6 Hz), 7.21 (m, 3H), 7.33 (t, 1H, *J* = 7.5 Hz), 7.45 (m, 3H), 7.57–7.69 (m, 5H); ^13^C NMR (75 MHz, CDCl_3_) δ: 21.5, 29.4, 43.0, 49.1, 113.4, 123.0, 123.9, 126.4, 126.5, 129.0, 129.4, 129.9, 130.5, 130.8 (2C), 131.0, 131.4, 135.4, 137.1, 140.9, 143.0, 161.8, 194.3. HRMS (ESI, *m*/*z*) [M + H]^+^ calcd. for [C_28_H_26_BrN_2_O_2_]^+^ 501.1172; found 501.1159.

*N-(3-chlorobenzyl)-5-methyl-4-(4-methylbenzoyl)-1-(4-methylbenzyl)-1H-pyrrole-2-carboxamide* (**9h**) 

White solid. Yield: 78%, mp: 134–136 °C. ^1^H NMR (300 MHz, CDCl_3_) δ: 2.36 (s, 3H), 2.46 (s, 3H), 2.55 (s, 3H), 4.50 (d, 2H, *J* = 5.8 Hz), 5.72 (s, 2H), 6.26 (brt, 1H, *J* = 6.2 Hz), 6.87 (s, 1H), 6.96 (d, 2H, *J* = 7.5 Hz), 7.09–7.16 (m, 3H), 7.22–7.31 (m, 5H), 7.72 (d, 2H, *J* = 8.0 Hz); ^13^C NMR (75 MHz, CDCl_3_) δ: 12.2, 21.4, 21.9, 43.0, 48.2, 115.5, 120.4, 124.7, 126.1, 126.5, 127.9, 128.0, 129.2, 129.6, 129.7, 130.2, 134.5, 134.8, 137.3, 137.6, 140.7, 142.0, 142.6, 161.7, 192.1. HRMS (ESI, *m*/*z*) [M + H]^+^ calcd. for [C_29_H_28_ClN_2_O_2_]^+^ 471.1834; found 471.1821.

*N-(3-chlorobenzyl)-4-(4-methoxybenzoyl)-5-methyl-1-(4-methylbenzyl)-1H-pyrrole-2-carboxamide* (**9i**) 

White solid. Yield: 69%, mp: 136–138 °C. ^1^H NMR (300 MHz, CDCl_3_) δ: 2.32 (s, 3H), 2.50 (s, 3H), 3.87 (s, 3H), 4.48 (d, 2H, *J* = 5.9 Hz), 5.68 (s, 2H), 6.21 (brt, 1H, *J* = 5.7 Hz), 6.83 (s, 1H), 6.93 (m, 4H), 7.07 (m, 3H), 7.22 (m, 3H), 7.80 (d, 2H, *J* = 8.7 Hz); ^13^C NMR (75 MHz, CDCl_3_) δ: 12.1, 21.4, 43.0, 48.2, 55.8, 113.8, 115.3, 120.5, 124.7, 126.1, 126.5, 128.0 (2C), 129.8, 130.3, 131.8, 132.8, 134.6, 134.8, 137.3, 140.7, 141.6, 161.8, 162.9, 191.2. HRMS (ESI, *m*/*z*) [M + H]^+^ calcd. for [C_29_H_28_ClN_2_O_3_]^+^ 487.1783; found 487.1770.

*1-(4-aminobenzyl)-4-benzoyl-N-(3-chlorobenzyl)-5-methyl-1H-pyrrole-2-carboxamide* (**10a**)

To a solution of EtOH (30 mL), THF (7 mL), and saturated solution of NH_4_Cl (7 mL), **9j** (0.305 g, 0.625 mmol) and iron powder (0.523 g, 9.37 mmol) were added, and the resulting mixture was stirred for 2 h at 100 °C. After cooling to room temperature, the reaction mixture was diluted with EtOH and filtered on a celite pad. The organic solvents were removed under vacuum and the residue was taken up in EtOAc, washed with brine, dried (Na_2_SO_4_), and filtered. The organic solvent was removed under reduced pressure, and the residue was purified by flash chromatography (SiO_2_) using EtOAc/petroleum ether (40–60 °C)/DCM (7: 1.5: 1.5) as eluent to provide the title compound as a white solid. Yield: 81%, mp: 129–131 °C. ^1^H NMR (400 MHz, CDCl_3_) δ: 2.54 (s, 3H), 3.57 (brs, 2H), 4.46 (d, 2H, *J* = 6.0 Hz), 5.57 (s, 2H), 6.24 (brt, 1H, *J* = 6.0 Hz), 6.60 (d, 2H, *J* = 8.2 Hz), 6.78 (s, 1H), 6.86 (d, 2H, *J* = 8.2 Hz), 7.06 (m, 1H), 7.21 (m, 2H), 7.41–7.53 (m, 4H), 7.75 (d, 2H, *J* = 7.7 Hz); ^13^C NMR (101 MHz, CDCl_3_) δ: 12.3, 43.0, 48.0, 115.6 (2C), 120.2, 125.0, 126.1, 127.4, 127.9, 128.0, 128.1, 128.5, 129.4, 130.3, 131.8, 134.8, 140.5, 140.8, 142.1, 146.0, 161.9, 192.3. HRMS (ESI, *m*/*z*) [M + H]^+^ calcd. for [C_27_H_25_ClN_3_O_2_]^+^ 458.1630; found 458.1624.

*(±)N-(3-chlorobenzyl)-4-(hydroxy(phenyl)methyl)-5-methyl-1-(4-methylbenzyl)-1H-pyrrole-2-carboxamide* (**10b**)

To a solution of **1** (0.300 g, 0.656 mmol) in EtOH/H_2_O (10:1.5), NaBH_4_ (0.027 g, 0.723 mmol) was added over 20 min. The reaction mixture was stirred at 50 °C for 18 h. After cooling to room temperature, the reaction mixture was taken up in EtOAc and washed with 1N HCl (2 × 20 mL). The organic phase was dried (Na_2_SO_4_) and filtered, and solvent was removed under vacuum. The resulting residue was purified by flash chromatography on silica gel using petroleum ether (40–60 °C)/EtOAc (7:3) as eluent to provide the title compound as a white solid. Yield: 78%, mp: 107–109 °C. ^1^H NMR (400 MHz, CDCl_3_) δ: 2.00 (d, 1H, *J* = 3.7 Hz), 2.19 (s, 3H), 2.31 (s, 3H), 4.38–4.50 (m, 2H), 5.62 (q, 2H, *J* = 16.0 Hz), 5.84 (d, 1H, *J* = 3.6 Hz), 6.08 (brt, 1H, *J* = 6.1 Hz), 6.42 (s, 1H), 6.88 (d, 2H, *J* = 7.8 Hz), 7.09 (d, 3H, *J* = 7.5 Hz), 7.18–7.30 (m, 4H), 7.34–7.42 (m, 4H); ^13^C NMR (101 MHz, CDCl_3_) δ: 10.7, 21.4, 42.9, 48.2, 70.0, 110.8, 123.9, 124.6, 126.0, 126.4, 126.5, 127.6, 127.8, 128.0, 128.7, 129.6, 130.2, 133.2, 134.8, 135.7, 136.9, 141.1, 142.2, 162.0. HRMS (ESI, *m*/*z*) [M + H]^+^ calcd. for [C_28_H_27_ClN_2_O_2_Na]^+^ 481.1653; found 481.1635.

### 3.2. Cell Cultures

Human epithelial melanoma A375 cells, human colorectal carcinoma HCT-116 cells, and normal rat L6 myoblast were grown in DMEM (Invitrogen, Paisley, UK) supplemented with 10% fetal bovine serum (FBS, Cambrex, Verviers, Belgium), L-glutamine (2 mM, Sigma, Milan, Italy), penicillin (100 units/mL, Sigma), and streptomycin (100 μg/mL, Sigma) and cultured in a humidified 5% carbon dioxide atmosphere at 37 °C, according to ATCC recommendations. A375 and HCT-116 were used as preclinical human cancer models in vitro, while rat L6 were used as control healthy cells.

### 3.3. Bioscreens In Vitro for Anticancer Activity

The antiproliferative activity of **1**, **9a**–**i**, and **10a**,**b** was investigated through the estimation of a “cell survival index”, arising from the combination of cell viability evaluation with cell counting, as previously reported [21]. The cell survival index is calculated as the arithmetic mean between the percentage values derived from the MTT assay and the automated cell count. Cells were inoculated in 96-microwell culture plates at a density of 104 cells/well and allowed to grow for 24 h. The medium was then replaced with fresh medium and cells were treated for an additional 48 h with a range of concentrations (from 1.5 to 25 μM) of **1**, **9a**–**i**, and **10a**,**b**. Using the same experimental procedure, cell cultures were also incubated with DMSO as negative controls (vehicle), as well as with cisplatin (cDDP) and nutlin-3a as positive controls for cytotoxic and antiproliferative effects, respectively. Cell viability was evaluated using the MTT assay procedure. Cell number was determined by TC20 automated cell counter (Bio-Rad, Milan, Italy), providing an accurate and reproducible total count of cells and a live/dead ratio in one step by a specific dye (trypan blue) exclusion assay. The calculation of the concentration required to inhibit the net increase in the cell number and viability by 50% (IC_50_) is based on plots of data (*n* = 6 for each experiment) and repeated five times (total *n* = 30). IC_50_ values were calculated from a dose–response curve by nonlinear regression using a curve-fitting program, GraphPad Prism 5.0, and are expressed as mean values ± SEM (*n* = 30) of five independent experiments.

### 3.4. Apoptosis/Necrosis Detection

The evaluation of apoptosis and/or necrosis induction after treatments in vitro was investigated by using the Apoptosis/Necrosis Detection kit (ab176749, Abcam, Discovery Drive, Cambridge Biomedical Campus, Cambridge, UK), which is optimized to simultaneously detect cell apoptosis (green), necrosis (red), and healthy cells (blue) by fluorescence microscopy analysis. During apoptosis, phosphatidylserine (PS) is transferred to the outer leaflet of the plasma membrane. As a universal indicator of the initial/intermediates stages of cell apoptosis, the appearance of PS on the cell surface can be detected before cytomorphological changes are observed. Thus, the phosphatidylserine (PS) early apoptotic sensor has green fluorescence (Ex/Em = 490/525 nm) upon binding to membrane PS. Conversely, loss of plasma membrane integrity, as demonstrated by the ability of a membrane-impermeable 7-AAD (Ex/Em = 546/647 nm) to label the nucleus, represents a straightforward approach to demonstrate late-stage apoptosis as well as necrosis. In addition, this kit also provides a live-cell cytoplasm-labeling dye, CytoCalcein Violet 450 (Ex/Em = 405/450 nm), for labeling living-cell cytoplasm. A375 and HCT-116 cells were inoculated in 96-microplates (black wells/clear flat bottom) at a density of 5 × 103 cells × well and allowed to grow for 24 h. Then, cells were treated for 24 h with IC_50_ values of **9c** and **10a** for A375 and of compound **9f** and **(±)10b** for HCT-116 cells, respectively. Cells were also incubated with nutlin-3a, here used as positive controls. After removing the medium, the cells were washed three times with 100 µL assay buffer, stained with 200 µL assay buffer, 2 µL Apopxin Green Solution, 1 µL 7-ADD, and 1 µL CytoCalcein Violet 450, and incubated at RT for 30 min protected from light. Finally, 200 μL assay buffer was replaced and cells were analyzed under a fluorescence microscope at Ex/Em = 490/525 nm for apoptosis detection, at Ex/Em = 550/650 nm for necrosis detection, and at Ex/Em = 405/450 nm for healthy cell recognition.

### 3.5. Statistical Analysis

All data were presented as mean values ± SEM. The statistical analysis was performed using Graph-Pad Prism (Graph-Pad software Inc., San Diego, CA, USA), and an ANOVA test for multiple comparisons was performed followed by Bonferroni’s test.

### 3.6. Solubility Assay 

The solubility of each molecule in PBS was quantified measuring the UV scattering in the range of 600–800 nm (JASCO UV-530 spectrophotometer, equipped with ETC-505 T temperature controller). DMSO solutions of **1**, **9a**, **9b**, **9c**, **9d**, **9e**, **9f**, **±10b**, and nutlin-3a were used at a final concentration of 2.6 mM, whereas those of **9g**, **9h**, and **9i** were used at 1.0 mM. A total of 1400 µL of degassed PBS was placed in a quartz cuvette (o.l. 0.5 cm), equilibrated 5 min at 37 °C, and then titrated by 7–10 successive additions of 0.5 µL or 0.3 µL from the DMSO solution. Each titration was performed without removing the cuvette from the UV cell and, after each addition, the solution was equilibrated 1 min a 37 °C under gently stirring before the acquisition of the UV data. For each molecule, the solubility was expressed as the mean ± SEM (*n* = 3).

### 3.7. Molecular Modeling Studies 

Molecular modeling calculations were performed on SGI Origin 200 8XR12000 and E4 Server Twin 2 × Dual Xeon 5520 equipped with two nodes. Each node: 2 × Intel Xeon QuadCore E5520, 2.26 Ghz, 36 GB RAM. The molecular modeling graphics were carried out on a personal computer equipped with an Intel(R) Core (TM) i7-4790 processor and SGI Octane 2 workstations. 

#### 3.7.1. Conformational Analysis

The apparent pKa and clogD (pH 7.4) values of **9a**–**i** and **10a**,**b** were calculated by using ACD/Percepta software (ACD/Percepta software, version 2017.1.3, Advanced Chemistry Development, Inc., Toronto, ON, Canada, 2017, http://www.acdlabs.com, accessed on 5 September 2022). All new compounds and the reference compound **1** were considered neutral in all calculations performed as a consequence of the estimation of percentage of neutral/ionized forms computed at the pH 7.2 (cytoplasmic value) using the Handerson–Hasselbalch equation. The newly designed pyrrole derivatives were built using the Small Molecule tool of Discovery Studio 2017 (Dassault Systèmes BIOVIA, San Diego, CA, USA). Atomic potentials and charges were assigned using the CHARMm force field [22]. Then, all compounds were subjected to molecular mechanic (MM) energy minimization (ε = 80 * r) until the maximum RMS derivative was less than 0.01 kcal/Å using Conjugate Gradient as minimization algorithm [23].

All the rotatable bonds included in the conformational search are sigma bonds not included in rings and, by consequence, are free to rotate. Nevertheless, the torsion angles τ1, τ2, and τ3 are affected by π electron conjugation and can be found in a synperiplanar or antiperiplanar conformation. In any case, it is known that the transition occurs in solution and that the two isomers can be detected by NMR lowering the temperature [24,25]. 

The conformational space of the compounds was sampled using the random search algorithm Boltzmann Jump for the random generation of a maximum of 400 conformations. Applying this method, each random perturbation is either accepted or rejected according to the Metropolis selection criterion with a ratio according to the Boltzmann distribution (T = 300 K). Finally, an energy threshold value of 20 kcal/mol was used as selection criteria. The generated structures were then subjected to MM energy minimization until the maximum RMS derivative was less than 0.01 kcal/Å using Conjugate Gradient [23] as minimization algorithm and the Generalized Born implicit solvent model with a solvent dielectric constant value of 80 [26]. The resulting conformers were ranked by their potential energy values (i.e., energy difference from the global minimum (ΔE_GM_)) and those showing a ΔE_GM_ ≤ 5 kcal/mol were selected for further analysis.

The selected structures were classified into conformational families named TTT, TTC, TCC, TCT, TCG^−^, TCA^−^, and TCG^+^ according to the values of the torsion angles τ1, τ2, and τ3 (i.e., synperiplanar, = C; antiperiplanar = T; synclinal = G^−^; anticlinal = A^−^; synclinal = G^+^). The obtained families were divided into subfamilies according to the values of the torsion angle τ5 (using the roman numbers). The ranges of torsion angle values used for the classification were the following: 0° to ±30° (synperiplanar, C); 30° to 90° (synclinal, G^+^); −30° to −90° (synclinal, G^−^); 90° to 150° (anticlinal, A^+^); −90° to −150° (anticlinal, A^−^) and ±150° to 180° (antiperiplanar, T) [27].

The lowest-energy conformers of each conformational family were then subjected to DFT calculations (Gaussian 16 package) [28]. Since the compounds **1**, **9a**–**f**, **9h**–**i**, and **10a** presented—in the case of the conformational families TCC and TCT—a MM lowest-energy conformer belonging to two different subfamilies, we included in the DFT calculations of these compounds the minimum conformers of the subfamilies I and II for the TCC family and the minimum conformers of the subfamilies I and III for the TCT family. All structures were fully optimized at the B3LYP/6–31+G(d,p) level [29,30] using the conductor-like polarizable continuum model (C-PCM) [18]. The C-PCM method allows the calculation of the energy in the presence of a solvent. In this case, all structures were optimized as a solute in an aqueous solution. In order to characterize every structure as minimum, a vibrational analysis was carried out (keyword = freq). The RMS force criterion was set to 3 × 10^−4^ a.u. The electronic distribution has been calculated using the natural bond orbital (NBO) method [31]. 

The resulting DFT conformers were ranked by their potential energy values (i.e., ΔE from the global energy minimum) and those showing a ΔE_GM_ < 2 kcal/mol were selected and classified into conformational families/subfamilies by applying the same criteria used for the MM conformers. The distances between the pharmacophore moieties were calculated for each conformer using the centroids of the aromatic rings X, Y, and Z.

#### 3.7.2. Structural and Bioinformatics Studies 

The experimentally determined structures of (i) the two-stranded parallel coiled coil of the GCN4 leucine zipper, (PDB ID: 2ZTA); (ii) the antiparallel coiled coil of GreA transcript cleavage factor from *E. coli* (PDB ID: 1GRJ); and (iii) the LxxLL motif of the nuclear receptor coactivator 5 in complex with the estrogen receptor β (PDB ID: 2J7X) were downloaded from the Protein Data Bank (PDB; http://www.rcsb.org/pdb/, accessed on 12 December 2022). All structures were analyzed using the Simulation and Macromolecule tools of Discovery Studio 2017 (Dassault Systèmes BIOVIA, San Diego, CA, USA).

The centroids of the hot-spot residues responsible for PPI (i.e., coiled coil: i, i + 3, i + 4, i + 7; LxxLL: i, i + 3, i + 4) were built considering the ring carbon atoms of aromatic side chains and all the heavy atoms of aliphatic amino acids side chains. Then, the distances between the centroids of the interacting residues were calculated. The resulting distance maps were crossed with the pharmacophore distances of the TSPs, considering the lowest-energy DFT conformers (ΔE_GM_ ≤ 2 kcal/mol). According to the pharmacophoric distances (using the same specified distance similarity criteria), the selected subfamilies were grouped, and, for each group, a pharmacophore was built calculating the average values of the pharmacophore distances. The TSP conformational subfamilies showing the best match with the experimentally determined distances of the interacting residues of the PPI motifs were selected (two distances differing less than 1.0 Å and one less than 1.5 Å). 

A representative conformer of each pharmacophore (considering both conformational/configurational enantiomers) was superimposed on the PPI motifs by fitting the centroids of the TSP pharmacophore groups (i.e., the aromatic rings X, Y, and Z) on the centroids of the side chains of the interacting residues of the PPI motifs, and the root-mean-square distance (RMSD) values of the fitted conformers were calculated. The TSP conformers with a calculated root-mean-square distance (RMSD) value < 2.5 Å were considered mimetic of that motif.

A structural and bioinformatics analysis was then performed on the experimentally determined structures of p19^ARF^, a putative molecular target of our TSPs. The structure of mouse p19^ARF^ tumor suppressor protein (PDB ID: 1HN3) was downloaded from the Protein Data Bank (PDB; http://www.rcsb.org/pdb/, accessed on 12 December 2022). Moreover, the full-length sequence of the human homologue of mouse p19^ARF^ (p14^ARF^) was downloaded from the UniProtKB/Swiss-Prot Data Bank (http://www.uniprot.org, accessed on 12 December 2022). A pairwise alignment was performed using the sequences of p19^ARF^ and its human homologue p14^ARF^ (PROMALS 3D server; http://prodata.swmed.edu/promals3d/promals3d.php, accessed on 12 December 2022) [32]. 

The presence of protein-recognition motifs was checked on the first two N-terminal helices of p14^ARF^ (αH1, residues 4–14; αH2, residues 20–29) (consensus sequence: [VLIFYWM]xx[VLIFYWM][VLIFYWM]xx[VLIFYWM]; Predict Sequence Properties protocol, Discovery Studio 2017). A homology model of human p14^ARF^ was built (Macromolecule, Discovery Studio 2017) on the bases of the sequence alignment with mouse p19^ARF^, and the hot-spot residues of the identified PPI motifs on the αH2 of p14^ARF^ were introduced in the structure of p19^ARF^ (PDB ID: 1HN3). Finally, the selected DFT conformers of our TSPs matching the above-specified similarity criteria with the considered PPI motifs were superimposed on the newly identified PPI motif on human p14^ARF^ (H2) by fitting the centroids of the pharmacophore groups (i.e., the aromatic rings X, Y, and Z) on the centroids of the side chains of the hot-spot residues of the PPI motif. The RMSD values of the fitted conformers were calculated.

## 4. Conclusions

Starting from the previously identified hit candidate **1**, a small and rationally designed series of new anticancer TSP derivatives has been developed, leading to the identification of a new hit (**10a**) with improved solubility and the highest activity among the present and previous TSPs.

The combination of computational and experimental studies allowed us to refine our 3D pharmacophore model, identifying the role of the different substituents on the pharmacokinetics and pharmacodynamics of these compounds. The whole of our results supports the hypothesis that the anticancer activity of these TSP derivatives is related to their ability to reproduce structural motifs involved in the PPIs activating apoptotic cell-death pathways. In particular, our TSPs share some peculiar structural features with the hot-spot residues of protein motifs involved in the formation of parallel and antiparallel coiled coil structures, such as that present on the N-terminal domain of the tumor suppressor p14^ARF^. 

The results of the present studies form the bases for the design of future investigations aimed at the development of new anticancer agents against melanoma (p53 WT, CDKN2A mut.).

## Data Availability

The original data presented in the study are included in the article; further inquiries can be directed to the corresponding authors.

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
