# Peer review of "Tetrasubstituted Pyrrole Derivative Mimetics of Protein–Protein Interaction Hot-Spot Residues: A Promising Class of Anticancer Agents Targeting Melanoma Cells"

_molecules, 2023, doi:10.3390/molecules28104161_

Round 1

Reviewer 1 Report

The article «Tetrasubstituted pyrrole derivatives mimetics of protein-protein interaction hot spot residues: a promising class of anticancer agents targeting melanoma cells» by M. Persico et al. is related to the problem of developing new anti-melanoma agents. The subject of the paper concern such molecular design, synthesis and biological properties of tetrasubstitued derivatives of pyrrole. All experimental studies are scientifically valuable and their description is well-written. At the same time the computational studies can be improved to became more informative. First, authors should add the information about the rotation barrier around torsion angles τ to clarify the possibility of conformational changes in solution. Second, the method focused on superimposition of small molecule into protein chain is very rough. Authors can calculate the complexes of TSP with some fragments of protein chains to analyze their mutual influence. PM7 or GFN2-xtb methods (the most accurate among semi-empirical methods) can be used for this purpose with moderate computational effort. Third, the molecular recognition between TSP and protein chains can be analyzed with electrostatic potential function. In addition, biological activity of TSP can be predicted with such services like Pass online and then compared with experimental studies.

Thus, the paper can be published in Molecules after revision.

Author Response

Legenda: R: reviewer; A: answer

R. First, authors should add the information about the rotation barrier around torsion angles τ to clarify the possibility of conformational changes in solution.

A. All the rotatable bonds included in the conformational search (i.e., torsion angles τ) are sigma bonds not included in rings and, by consequence, subjected to conformational changes in solutions at room temperature. Nevertheless, the torsion angles t1, t2, and t3 are affected by π electron conjugation and can be found in a synperiplanar or antiperiplanar conformation. Due to steric effects, the coplanarity of the ground state is never found in di-aryl ketones (t2 and t3 ) as well as in amides conjugated with aromatic rings, thus affecting the torsional barrier of the syn/anti conformational transition. Going into details, dedicated studies based on spectroscopic data (e.g., Nakamura Nobuo and Oki Michinori, Restricted Rotation in Aromatic Ketones. I. Substituent Effects on the Barrier to Rotation about the Benzene-to-Carbonyl Bond Bulletin of the Chemical Society of Japan 1972 45:8, 2565-2570 - Ryan A. Olsen, Lisa Liu, Nima Ghaderi, Adam Johns, Mary E. Hatcher, and Leonard J. Mueller The Amide Rotational Barriers in Picolinamide and Nicotinamide:  NMR and ab Initio Studies Journal of the American Chemical Society 2003 125 (33), 10125-10132 DOI: 10.1021/ja028751j), reported that ortho-substituted aryl ketones (t2 and t3) have a torsional energy barrier value from 7 to 15 kcal/mol while in the case of conjugated amide bonds similar to that defined by t1, the torsional energy barrier value can reach  20 kcal/mol. In any case, it is known that the transition occurs in solution and that the two isomers can be detected by NMR lowering the temperature.

According to the reviewer observation, we clarified the possibility of the torsion angles t  to undergo conformational changes in solution by inserting this information in the manuscript at page 22 lines 798-803 together with two new references (24 and 25).

R. Second, the method focused on superimposition of small molecule into protein chain is very rough. Authors can calculate the complexes of TSP with some fragments of protein chains to analyze their mutual influence. PM7 or GFN2-xtb methods (the most accurate among semi-empirical methods) can be used for this purpose with moderate computational effort. Third, the molecular recognition between TSP and protein chains can be analyzed with electrostatic potential function.

A. We thank the reviewer for giving us the opportunity to clarify this issue and avoid any misunderstanding on the computational procedure applied in this work. Indeed, no attempt of producing TSP/protein complexes has been performed and, by consequence, of calculating the interactions involved in the molecular recognition between TSPs and the protein target.

The mentioned structural comparison (paragraph 2.5.2 Investigation of the peptidomimetic ability of the new TSPs) was not made to dock any structure but to identify the helix-based hydrophobic motif(s) whose consensus structure (i.e., the spatial orientation of the hot spot residues) can be mimicked by TSPs. Results obtained indicated the ability of TSPs to mimic the structural arrangement of I, i+3/i+4, i+7 residues on an a-helix (i.e., the hot spot residues of coiled coil motifs).

In particular, on the bases of the current and previous biological data as well as the results of a bioinformatics and structural analysis, we hypothesized that TSPs could mimic a coiled coil motif of the N-terminal domain of p14ARF. The N-terminal domain of p14ARF forms a complex with MDM2 by binding to a site other than that recognized by p53, however no structural data are available. Hence - to model the interaction of TSPs with the putative molecular target MDM2 - at first, we should model the p14ARF/MDM2 interaction. Therefore, no attempt was made in this direction, at least at this stage of our research.

According to reviewer observations, the manuscript has been modified at page 12 (lines 343-347)  and 14 (lines 442-444).

R. In addition, biological activity of TSP can be predicted with such services like Pass online and then compared with experimental studies.

A. We already explored, in this and previous works, the activity profile and the mechanism of action of TSPs and we especially did it on the bases of their structural similarity with the hot spot residues of protein motifs involved in the p53 pathway. Such a rationale is similar to the 2D-structural comparison performed by the algorithm provided by servers such as PASS Online, which, however, do not include neither 3-D structure comparison nor the comparison with protein motifs.

The main advantage of software such as PASS Online is to obtain the predicted biological activity profile using only structural formula so that the “prediction is possible even for virtual structure designed in computer but not synthesized yet”. In the present work, we did not indicate any new compound to be synthesized, accordingly, we did not include any qualitative or quantitative prediction of biological activity. It has to be underlined that, in any case, the prediction of reliable activity values (i.e., to be compared with experimental values) would be anyway hampered by the low variance and high SD of the IC50 currently available.

Reviewer 2 Report

The manuscript concerns the synthesis and biological activity of tetrasubstituted pyrrole derivatives that showed the ability to inhibit the growth of two different human cancer cell lines without any significant effect on non-cancerous cells. The synthetic route description is understandable and the characterisation of the final compounds is supported by analytical data. The best compounds demonstrated the ability to activate apoptotic cell death pathways. Furthermore, the authors well described the structure-activity relationship and the role of the new derivatives as peptidomimetics through in silico studies.

Author Response

We thank the reviewer for her/his positive comments on our work.

Reviewer 3 Report

The paper presents peptidomimetic tetrasubstituted pyrrole (TSP) derivatives that are potential anticancer agents targeting melanoma.

Authors designed the research carefully and showed the design and synthesis of TSP derivatives, solubility, biological assay, apoptosis studies, and peptidomimetic ability by molecular modeling.

Publication of the paper is recommended.

Author Response

(The authors gave the same response as above.)

Round 2

Reviewer 1 Report

Authors improved the quality of presentation of their results in new version, so now the article can be accepted for publication.